Geoscientific Model Development (gmd)

# An improved method of the Globally Resolved Energy Balance Model by the Bayesian Networks

Zhenxia Liu[1], Zengjie Wang[1], Jian Wang[1], Zhengfang Zhang[1], Dongshuang Li[4,5], Zhaoyuan Yu[1,2,3], Linwang Yuan[1,2,3], and Wen Luo[1,2,3]

[1]School of Geography, Nanjing Normal University, Nanjing, 210023, China
[2]Key Laboratory of Virtual Geographic Environment (Nanjing Normal University), Ministry of Education, Nanjing, 210023, China
[3]Jiangsu Center for Collaborative Innovation in Geographical Information Resource Development and Application, Nanjing, 210023, China
[4]Jiangsu Key Laboratory of Crop Genetics and Physiology/Jiangsu Key Laboratory of Crop Cultivation and Physiology, Agricultural College of Yangzhou University, Yangzhou, China
[5]Jiangsu Co-Innovation Center for Modern Production Technology of Grain Crops, Yangzhou University, Yangzhou, China,

**Correspondence:** Wen Luo (09415@njnu.edu.cn)

**Abstract.** The accurate simulation of climate is always critically important and also a challenge. This study introduces an improved method of the Globally Resolved Energy Balance Model (GREB) by the Bayesian Networks based on the concept of coarse-fine model. The improved method constructs a coarse-fine structure that combines a dynamical model with a statistical

model based on employing the GREB model as the global framework, and utilizing a Bayesian Networks constructed on the interrelationships between internal climate variables of the GREB model to achieve local optimization. To objectively validate the performance and generalization of the improved method, the method is applied to the simulation of surface temperature and temperature of the atmosphere based on the $3.75° \times 3.75°$ global data sets by Environmental Prediction (NCEP)/ National Center for Atmospheric Research(NCAR) from 1985 to 2014. The results demonstrate that the improved model exhibits higher

average accuracy and lower spatial differentiation than the original GREB model, and is robustness in long-term simulations. This approach addresses issues with the accuracy of the GREB model in local areas, which can be attributed to an over-reliance on boundary and initial conditions, and a lack of fully using observed data. Additionally, it overcomes the challenge of poor robustness in statistical models due to ambiguous climate inclusions. Thus, the improved method provides a promising way to give reliable and stable simulation of climate.

## 1   Introduction

As the global warming progresses, extreme events and meteorological disasters occur frequently(Grant, 2017). Thus, the simulation and prediction of climate have become an important topic in current scientific research for the conceptual understanding and development of hypotheses for climate change studies(Dommenget and Flöter, 2011; Huang et al., 2019). Climate models are mathematical models that describe the temporal evolution of climate, oceans, atmosphere, ice, and land-use processes,

across a spatial domain via systems of partial differential equations(Berrocal et al., 2012), which can be solved by supercomputer and is an important tool for simulating and predicting future climate change(Kay, 2020).

Generally, climate models mainly include two categories, dynamic model and statistical model. Dynamic model can well understand and express the dynamic process of climate by modeling various complex climate processes or interactions, but it still faces two major problems: i) The simulation process overly relies on initial conditions and boundary conditions(Alley et al., 2019; Zhang et al., 2019; Ludescher et al., 2021); ii) The climate model is too complicated, and its internal characteristics cannot be fully expressed(Fan et al., 2021; Zou et al., 2019; Feng et al., 2020). The Globally Resolved Energy Balance Model (GREB) is a simple but representative dynamic model, which is based on energy balance theory(Dommenget and Flöter, 2011). Compared with other dynamic models, the GREB model is a relatively fast tool for the conceptual understanding and development of hypotheses for climate change studies, because it computes about one model year per second on a standard personal computer, which allows conducting sensitivity studies to external forcing within minutes to hours(Dommenget and Flöter, 2011; Dommenget, 2016; Stassen et al., 2019). However, in addition to the two main problems of dynamic models, the GREB model also faces the problem that the model does not respond well to anomalous climate change because the parameters of the GREB model are predetermined and the observed data can hardly be used to dynamically correct the model parameters(Dommenget and Flöter, 2011; Dommenget, 2016). How to solve these problems is an important research topic to improve the GREB model and further extend it to other dynamic models.

On the contrary, statistical model, as another type of climate models, can make good use of historical observation data to dynamically modify the models from data(Feng et al., 2020), and solve the problem that dynamic climate models rely too much on initial and boundary conditions and underutilize full observation data. Therefore, it provides a possible way to solve those defects of the dynamical model by combining that with the statistical model. Bayesian Networks is a statistical method which combines graph theory and probability (Cai et al., 2013, 2019; Jansen et al., 2003). The method uses graph to express the structure relation of the variables related to the model and has the characteristics of structuring and quantifying the object relation through the causal relation among the parts of the probability computing system(Pearl, 1986), variable logic reasoning and predictive simulation can be realized, and it can use a large amount of historical observation data. As described, it is a possible way to improve the GREB model by the Bayesian Networks.

The concept of coarse-fine model provides a joint modeling approach of dynamical-statistical hybrid model that is different from the traditional use of statistical model to optimize the empirical parameters of the dynamical model. It starts from different coarse and fine granularity of the model(Akgul and Kambhamettu, 2003; Pal and Bhattacharya, 2010; Yibo et al., 2009), uses the dynamical model as a global framework and uses the statistical model to do local optimization, and realizes the unified modeling of both. Based on this idea, this paper introduces a method for improving the GREB model by the Bayesian Networks. The aim of method is to solve the problem of low model accuracy due to over-reliance on boundary conditions and initial conditions and inability to fully utilize historical observation data. The following section presents the improved method. Section 3 presents the study case and data sets to test the new improved model. Finally, we give a discussion and conclusion of the results.

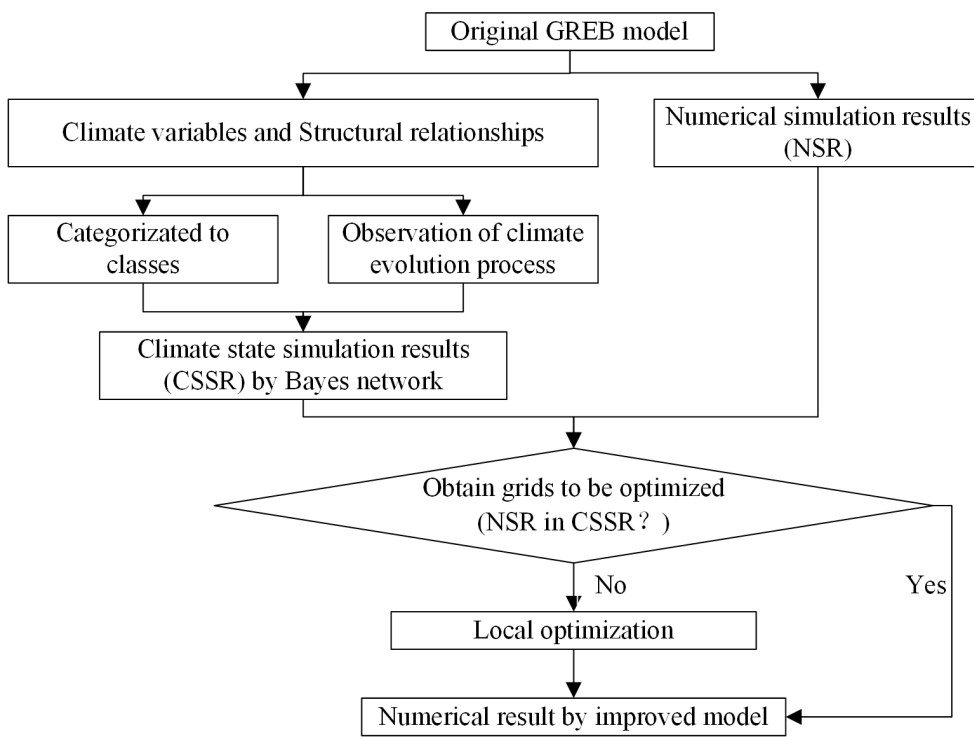

**Figure 1.** Overall framework of the improved method.

## 2   Methods

The improved method is developed according to the following procedure. Firstly, climate variables representing different climate processes are chosen as nodes in the Bayesian Networks constructed by the GREB model. And the structural relationships among different nodes are determined to establish an abstract model of the components and structural relationships of climate processes. Secondly, the selected climate variables are categorized into variable ranges based on their numerical values to form different classifications that are used to indicate different climate state. Thirdly, the climate state simulation method is

reconstructed based on the Bayesian Networks and climate evolution process to achieve the simulation of the target variables climate state. Finally, the climate state simulation results obtained from the Bayesian Networks are compared with the climate modal simulation results from the original GREB model to get the local optimization grids, and the numerical results of the original GREB model simulation are optimized based on the comparison results. Based on the above considerations, improved method is developed according to the following procedures (Figure 1).

### 2.1   Structural relationship among climate variables

Based on the energy balance, the GREB model can simulate the main characteristics and climate mean states of global warming, including seven climate processes (solar radiation, thermal radiation, hydrological cycle, sensible heat and atmospheric temper-

ature, atmospheric circulation, sea ice and deep ocean) and four main climatic variables (surface temperature, temperature of the atmosphere, temperature of subsurface the ocean, and humidity of the surface). Each of these processes is represented with strongly simplified equations. Therefore, we can abstract the structural relationship among different climate variables from the simplified equation, i.e., which climate variables control a given climate variable, and which climate variables are influenced by it. This structural relationship provides the possibility to construct Bayesian Networks.

## 2.2 Categorization of climate state

According to the theory of climate sensitivity(Annan and Hargreaves, 2006; Dommenget, 2016), Climate state, indicated by a range of numerical values, can be used to replace the specific numeric to simulate climate change and characterize the long-term trend of climate change and extreme weather conditions. And it is better suited to capture the similarity of a given climate variable across different spatial and temporal locations compared to the numerical values of specific climate variables. Therefore, it can be used to assess the similarity between the simulated results of a model and the actual results, indicating the accuracy of the simulation. This provides a simple and practical approach to evaluating the accuracy of revealing local abrupt changes in simulation results. Moreover, by simulating state rather than specific numeric, it is possible to significantly reduce computational effort and simulation response time. This is consistent with the primary objective of the GREB model, which is to provides a fast tool for the conceptual understanding and development of hypotheses for climate change studies(Dommenget and Flöter, 2011)

The natural breaks classification (Jenks) method is a commonly used classification method that aims to minimize intra-class variation and maximize inter-class variation. By categorizing the numeric of climate variables into different classifications to indicate climate state using the natural breaks classification method, it can be considered as that numeric within the same classification have less variation, representing that the results of this classification of numeric have similar climate state.

## 2.3 Climate state simulation

According to the characteristics of Bayesian Networks, the climate state simulation of climate variables is realized by climate evolution process based on Bayesian Networks, i.e., the climate state of unknown climate variables is inferred from the climate state of known climate variables at the same spatial locations.

### 2.3.1 Bayesian Networks

Bayesian Networks is a probabilistic model that simulates the human reasoning process, which is a combination of graph theory and probability theory, and its network topology is a directed acyclic graph. Where variables are nodes and correlations or causal relationships between variables are directed edges.The dynamic evolution of Bayesian Networks node probabilities is controlled by conditional probabilities, and each node covers a probability distribution table under the joint distribution of the parent nodes, indicating the strength of the relationship between the nodes.(Sahin et al., 2019). When the Bayesian Networks is constructed, given the state of any node, the probability distribution of the states of the remaining nodes can be calculated.

In the Bayesian Networks, the probability of a node can be calculated in the form of probability using prior knowledge and statistical data, namely the Bayes probability (Maher, 2010). Observed sample are defined as: $G = \{X_1 = x_1, X_2 = x_2, \cdots, X_n = x_n\}$, where $X$ is event, $x$ is event value or state. When $\theta$ is the prior probability of event $X = x$, $\zeta$ is prior knowledge, $P(\theta|\zeta)$ is probability density function, then the probability $P(X_{n+1} = x_{n+1}|\theta,\zeta)$ of the $n+1$ event $X_{n+1} = x_{n+1}$ can be obtained from the prior probability density $P(\theta|\zeta)$ and the sample $G$ through the Bayes probability. It can be calculated by total probability formula:

$$
\begin{aligned}
&P(X_{n+1} = x_{n+1}|\theta,\zeta) \\
&= \int P(X_{n+1} = x_{n+1}|\theta,\text{G},\zeta)P(\theta|\text{G},\zeta)\,d\theta \\
&\qquad = \int \theta P(\theta|\text{G},\zeta)\,d\theta
\end{aligned}
\tag{1}
$$

Based on Bayes equation, The posterior probability $P(\theta|\text{G},\zeta)$ is denated as:

$$
P(\theta|\text{G},\zeta) = \frac{P(\theta|\zeta)\,P(\text{G}|\theta,\zeta)}{P(\text{G}|\zeta)}
\tag{2}
$$

Where $G$ is given sample, $\zeta$ is priori probability of $G$,

### 2.3.2 Climate evolution process based on Bayesian Networks

In a climatic process composed of several climatic variables, there is an association relationship between climatic variables. These climatic variables are regarded as network nodes, and the association relations between climatic variables are taken as directed edges. The association relationship between nodes is represented by graph model, and the action intensity of association relationship is described quantitatively by conditional probability table. Using the characteristics of Bayesian Networks, the attribute feature state of nodes is inferred by probability. To realize the expression and simulation of the attribute feature state of geographical variables.

A climate process $M_t = \{X(m_1, m_2, \ldots, m_i)|m_{1t}, m_{2t}, \ldots, m_{it}\}$ is composed with $i$ climate variables, $m_1, m_2, \ldots, m_i$, and $X(m_1, m_2, \ldots, m_i)$ is the structural relationship among the variables. Suppose that the climate variable $m_i$ has $j$ states, then the sataes set of $m_i$ is $\{W_{m_{i1}}, W_{m_{i2}}, \ldots, W_{m_{ij}}\}$. The climate process is described by a Bayesian Networks $B = (S, X)$. where $S$ is a directed acyclic graph composed of nodes; $X$ is the nodes set of graph, that is climate variables $m_1, m_2, \ldots, m_i$. nodes are connected by directed edges to represent the relationship between climate variables. Each node has an independent conditional probability table, which represents the probability distribution under the joint distribution of its parent nodes. Assume that a climate $m_i$ has one or more parent nodes $m_1, m_2, \ldots, m_e (e \leq i - 1)$ and states $d_1, d_2, \ldots d_e$, it can be denoted as : $m_1, m_2, \ldots, m_e \to m_i$. Under the parent node of all possible states, the conditional probability table composed of the set of state probabilities of mdeteci is follow:

$$
\begin{aligned}
&B_{m_i}^{W_{m_1 r_1}, W_{m_2 r_2}, \ldots, W_{m_e r_e}} = \\
&\left\{(W_{m_{i1}}, P_{W_{m_{i1}}}^{W_{m_1 r_1}, W_{m_2 r_2}, \ldots, W_{m_e r_e}}), (W_{m_{i2}}, P_{W_{m_{i2}}}^{W_{m_1 r_1}, W_{m_2 r_2}, \ldots, W_{m_e r_e}}), \ldots, (W_{m_{ij}}, P_{W_{m_{ij}}}^{W_{m_1 r_1}, W_{m_2 r_2}, \ldots, W_{m_e r_e}})\right\} \\
&\qquad (r_1 = 1, 2, \ldots, d_1), (r_2 = 1, 2, \ldots, d_2), \ldots, (r_e = 1, 2, \ldots, d_e)
\end{aligned}
\tag{3}
$$

Where $B_{m_i}^{W_{m_1r_1},W_{m_2r_2},...,W_{m_ere}}$ is a conditional probability table of climate variables $m_i$; $W_{m_{ij}}$ is $j$th characteristic state of climate variables $m_i$; $P_{W_{m_{ij}}}^{W_{m_1r_1},W_{m_2r_2},...,W_{m_ere}}$ is the probability of climate variable $m_i$ corresponds to the $j$th state under the $r1, r2, ... re$ characteristic state corresponding to the parent node $m_1, m_2, ..., m_e$ expression set. The probability set of climate variable $m_i$ at $t$ moment can be denoted as $C_{m_{it}}$:

$$
\begin{aligned}
C_{m_{it}} = \\
\left\{ (W_{m_{i1}}, P_{W_{m_{i1}}}^{W_{m_1r_1},W_{m_2r_2},...,W_{m_ere}}), (W_{m_{i2}}, P_{W_{m_{i2}}}^{W_{m_1r_1},W_{m_2r_2},...,W_{m_ere}}), ..., (W_{m_{ij}}, P_{W_{m_{ij}}}^{W_{m_1r_1},W_{m_2r_2},...,W_{m_ere}}) \right\}
\end{aligned}
\tag{4}
$$

The conditional probability table of each node can be calculated by Eq.2 using training data.

## 2.4 Local optimization

The numerical results simulated by the original GREB model are compared with the climate state results simulated by the Bayesian Networks, and the grids where the numerical result simulated by the original GREB model are not in the range of the climate state simulated by the Bayesian Networks are used as grids to be optimized.

According to the Third Law of Geography(Zhu et al., 2018), the more similar the geographic environment, the more similar the geographic target characteristics are. Therefore, for an unknown climate variable at a certain spatial and temporal location, the numeric of other known climate variables at that spatial and temporal location can be used to infer. Accordingly, we propose that for an unknown climate variable, the position of its specific value in the range of its classification is related to the position of the specific value of the known climate variable in the range of its classification at the same spatial and temporal location. For a climate variable containing $n$ relevant control variables, the numerical results are calculated as follows:

$$
E_{value}^x = S_{\text{lowerlimit}}^E + \frac{1}{n}(S_{\text{upperlimit}}^E - S_{\text{lowerlimit}}^E) \sum \frac{E_{value}^i - S_{\text{lowerlimit}}^i}{S_{\text{upperlimit}}^i - S_{\text{lowerlimit}}^i}
\tag{5}
$$

Where $E_{value}^x$ represents an unknown climate variable; $S_{\text{lowerlimit}}^E$ represents the lower limit of the range of classification in which the unknown climate variables are simulated by Bayesian Networks; $S_{\text{upperlimit}}^E$ represents the lower limit of the range of classification in which the unknown climate variables are simulated by Bayesian Networks; $n$ represents the number of known climate variables associated with the unknown variables in the Bayesian Networks; $E_{value}^i$ the actual value of the $i$th known climate variable; $S_{\text{lowerlimit}}^i$ represents the lower limit of the range of classification in which the $i$th known climate variables; $S_{\text{upperlimit}}^i$ represents the upper limit of the range of classification in which the $i$th known climate variables.

According to the above method, we can improve the accuracy of the model by comparing the climate state, identifying the grid to be optimized, and recalculate the values simulated by the original GREB model within the grid. In this way, the improved model with coarse-fine structure constructed with the GREB model as the global framework and the Bayesian Networks as the local optimization can better reflect the localized abrupt changes in the climate process and achieve the purpose of improving the GREB.

## 3 Case study

In order to demonstrate the accuracy of the improved model in simulating climate variables and to verify its reliability, surface temperatures and temperature of the atmosphere from the GREB model were selected for simulation objects. The simulation of these two climate variables includes most of the climate processes of the GREB and can reflect the complex coupling process and climate change characteristics of the GREB model.

### 3.1 Data description

In this paper, data produced by National Centers for Environmental Prediction (NCEP) / National Center for Atmospheric Research (NCAR) is used as the experimental data to evaluate the improved model. The data sets include surface temperature($T_{surf}$), temperature of the atmosphere($T_{atmos}$), solar radiation($F_{solar}$), total cloud cover($CLD$), water vapor ($q_{air}$), temperature of the subsurface ocean ($T_{ocean}$), and wind speed($\vec{u}$) stored as a 3.75° *3.75°(latitude * longitude) grid NC data from 1985 to 2014. In order to facilitate calculation and comparative analysis, all climate data is preprocessed. Firstly, the downloaded climate data is removed from the outliers so that the data are calculated to avoid too large or too small results; secondly, the grid data is resampled and the resampling method is bilinear interpolation. The bilinear interpolation method is used to interpolate the climate data, which not only fills the null values, but also unifies the scale size of the data. Finally, considering that changes in climate variables are usually seasonally related, climate data from 1985 to 2014 were processed as quarterly averages, where January, February, and March comprised first quarter, April, May, and June formed second quarter, July, August, and September constituted a third quarter, and October, November, and December comprised the fourth quarter.

### 3.2 Structural relationship among climate variables and climate state

The process of simulating the surface temperature includes solar radiation, thermal radiation, sensible heat and atmospheric temperature, and deep oceanDommenget and Flöter (2011). The main heat source of the surface temperature is solar radiation, some of which is absorbed by the surface temperature, the other part is reflected by the surface temperature, and part of the heat on the surface temperature is transferred in the atmosphere, and some of it is transferred to the ocean below the surface. Each climate variables in this scene can be expressed by a highly simplified equation, which follows the surface temperature tendency equation as follows:

$$\gamma_{surf}\frac{dT_{surf}}{dt} = F_{solor} + F_{thermal} + F_{latent} + F_{sense} + F_{ocean} \tag{6}$$

Where $T_{surf}$ is surface temperature; $\gamma_{surf}$ is surface heat capacity; $F_{solar}$ is the incoming solar radiation; $F_{thermal}$ is the net thermal radiation; $F_{latent}$ the cooling by latent heat from surface evaporation of water; $F_{sense}$ is the turbulent heat exchange with the atmosphere; $F_{ocean}$ is the heat exchange with the deeper subsurface ocean. The subprocesses of surface temperature

are modeled as follows:

$$
\begin{cases}
F_{solar} = (1-\alpha_{clouds})(1-\alpha_{surf})S_0 \cdot r(\phi, t_{julian}) \\
F_{thermal} = -\sigma T_{surf}^4 + \varepsilon_{atmos}\sigma T_{atmos-rad}^4 \\
\varepsilon_{atmos} = \frac{pe_8 - CLD}{pe_9} \cdot (\varepsilon_0 - pe_{10}) + pe_{10} \\
F_{latent} = L \cdot \rho_{air} \cdot C_w \cdot \left|\vec{u}_*\right| \cdot \upsilon_{soil} \cdot (q_{air} - q_{sat}) \\
F_{sense} = c_{atmos} \cdot (T_{atmos} - T_{surf}) \\
F_{ocean} = Fo_{sense} + \gamma_{surf} \cdot \Delta T_{entrain} \\
Fo_{sense} = c_{ocean} \cdot (T_{ocean} - T_{surf})
\end{cases} \tag{7}
$$

Where $F_{solar}$ is the incoming solar radiation; $\alpha_{clouds}$ is the fraction of the incoming solar radiation is reflected by clouds; $\alpha_{surf}$ is the fraction of the incoming solar radiation is reflected by the surface; $S_0$ is the solar constant; $\alpha_{surf}$ is the fraction of the incoming solar radiation is reflected by the surface; $r$ is the 24 h mean fraction reaching a normal surface area on top of the atmosphere; $\phi$ is the function of latitude; $t_{julian}$ the Julian day of the calendar year; $F_{thermal}$ is the net thermal radiation; $T_{surf}$ is surface temperature; $\varepsilon_{atmos}$ is the effective emissivity; $T_{atmos-rad}$ is the temperature defined in the context of the atmospheric temperature; $CLD$ is the total cloud cover; $\varepsilon_0$ is the emissivity without considering clouds first; $pe_*$ is the parameters; $F_{latent}$ the cooling by latent heat from surface evaporation of water; $L$ is the constant parameters of the latent heat of evaporation and condensation of water; $\rho_{air}$ is the density of air; $C_w$ is the transfer coefficient; $\left|\vec{u}_*\right|$ is the wind speed; $\upsilon_{soil}$ is the Bulk formula is extended by a surface wetness fraction; $q_{air}$ is the actual surface air layer humidity; $q_{sat}$ is the saturation surface air layer specific humidity; $F_{sense}$ is the turbulent heat exchange with the atmosphere; $c_{atmos}$ is the coupling constant; $T_{atmos}$ is temperature of the atmosphere; $F_{ocean}$ is the heat exchange with the deeper subsurface ocean; $Fo_{sense}$ is the turbulent mixing between the two ocean layers; $\Delta T_{entrain}$ is the heat exchange with the surface ocean layer due to decreasing of the mixed layer depth; $c_{ocean}$ is the coupling constant; $T_{ocean}$ is the temperature of the subsurface ocean.

In the process of simulating the temperature of the atmosphere by the GREB modelDommenget and Flöter (2011), temperature of the atmosphere is not only related to the thermal radiation reflected from the surface, but also related to the sensible heat exchange with the surface and latent heat release by condensation of atmospheric water vapor. Each climate variables in this process can be expressed by a highly simplified equation, which follows the temperature of the atmosphere tendency equation as follows:

$$
\gamma_{atmos}\frac{dT_{atmos}}{dt} + F_{sense} = F_{thermal} + Q_{latent} + \gamma_{atmos}\left(k \cdot \nabla^2 T_{atmos} - \vec{u} \cdot \nabla T_{atmos}\right) \tag{8}
$$

Where $T_{atmos}$ is temperature of the atmosphere; $\gamma_{atmos}$ is atmospheric heat capacity; $F_{sense}$ is the sensible heat exchange with the surface; $Fa_{thermal}$ is net thermal radiation of the atmosphere; $Q_{latent}$ is the latent heat release by condensation of atmospheric water vapor; $\vec{u}$ is the wind speed. The subprocesses of temperature of the atmosphere are modeled as follows:

$$
\begin{cases}
F_{sense} = c_{atmos} \cdot (T_{atmos} - T_{surf}) \\
Fa_{thermal} = \varepsilon_{atmos}\sigma \cdot T_{surf}^4 - 2\varepsilon_{atmos}\sigma \cdot T_{atmos-rad}^4 \\
Q_{latent} = -2.6736 \cdot 10^3 \left[kg/m^2\right] \cdot \Delta q_{precip} \cdot L
\end{cases} \tag{9}
$$

Where $F_{sense}$ is the turbulent heat exchange with the atmosphere; $c_{atmos}$ is the coupling constant; $T_{atmos}$ is temperature of the atmosphere; $Fa_{thermal}$ is net thermal radiation of the atmosphere; $T_{surf}$ is surface temperature; $\varepsilon_{atmos}$ is the effective emissivity; $T_{atmos-rad}$ is the temperature defined in the context of the atmospheric temperature; $Q_{latent}$ is the latent heat release by condensation of atmospheric water vapor; $\Delta q_{precip}$ is the condensation or precipitation; $L$ is the constant parameters of the latent heat of evaporation and condensation of water.

For different climate process, the climate subprocesses and relationship structures are different. Therefore, the selection of nodes in each climate process will also be different. Not only the selection of appropriate variables as nodes is very important, but also the number of nodes will directly affect the simulation of the final climate average state. In order to simplify the complex climate evolution process and facilitate calculation, 4-6 climate variables are selected as key nodes in each climate process, and the variables climate state in each processes are simulated by these nodes.

Through the trend equations (Eq. 6 and 7) in the processes of surface temperature, the relation equation of climate variables can be simplified:

$$
\begin{cases}
T_{surf} = f(F_{solar}, T_{ocean}, q_{air}, CLD) \\
T_{ocean} = f(F_{solar}) \\
q_{air} = f(F_{solar})
\end{cases}
\tag{10}
$$

Where $T_{surf}$ is surface temperature; $F_{solar}$ is solar radiation; $T_{ocean}$ is temperature of the subsurface ocean; $q_{air}$ is the actual surface air layer humidity, i.e., water vapor content; $CLD$ is total cloud cover. That is, surface temperature, solar radiation, temperature of the subsurface ocean, total cloud cover and water vapor content can be selected as the key nodes of the surface temperature process.

Through the trend equation (Eq. 8 and 9) in the processes of temperature of the atmosphere, the relation equation of climate variables can be simplified:

$$
\begin{cases}
T_{atmos} = f(\vec{u}, q_{air}, CLD) \\
q_{air} = f(\vec{u})
\end{cases}
\tag{11}
$$

Where $T_{atmos}$ is temperature of the atmosphere; $\vec{u}$ is wind speed; $CLD$ is total cloud cover; $q_{air}$ is water vapor content. That is, temperature of the atmosphere, wind speed, total cloud cover and water vapor content can be selected as the key nodes of the temperature of the atmosphere process.

According to Eq. 10 , in the surface temperature process, surface temperature ($T_{surf}$) is controlled by solar radiation ($F_{solar}$), cloud cover ($CLD$), water vapor ($q_{air}$) and temperature of the subsurface ocean ($T_{ocean}$). Temperature of the subsurface ocean ($T_{ocean}$) and water vapor ($q_{air}$) are controlled by solar radiation ($F_{solar}$). For the above relationship, the Bayesian Networks structure in the surface temperature process can be constructed (Figure 2a). According to Eq. 11, in the temperature of the atmosphere process, temperature of the atmosphere ($T_{atmos}$) is controlled by cloud cover ($CLD$), water vapor ($q_{air}$) and wind speed ($\vec{u}$). And water vapor ($q_{air}$) is controlled by wind speed ($\vec{u}$). For the above relationship, the Bayesian Networks structure in the temperature of the atmosphere process can be constructed (Figure 2b).

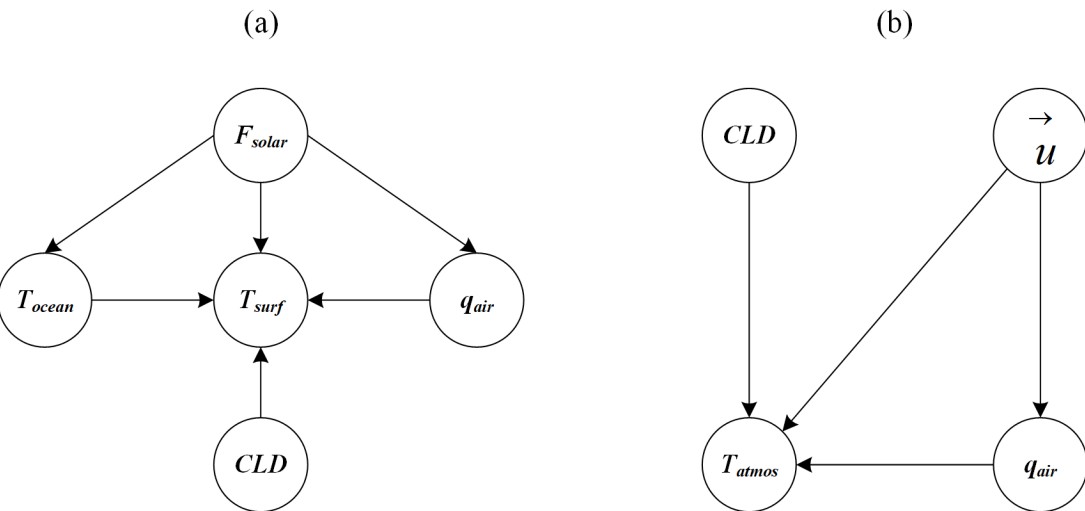

**Figure 2.** Structural relationship among climate variables in the different simulation processes. (a) Surface temperature process; (b) Temperature of the atmosphere process.

The climate state of the variables in the above climate processes was performed using the natural breaks classification method. The climate variables data are categorized into five, seven and nine different classifications to indicated different climate state to test the improved model and verify the effect of the classification number of climate variables data on the simulation results. Detailed schemes are shown in Appendix Table A1, A2, A3.

### 3.3 Climate state simulation

Surface temperature and temperature of the atmosphere are considered as a simulation object, and other climate variables as known objects, and uses the historical data to calculate the conditional probability tables of each nodes through the Bayesian Networks structure with Eq 4. Among them, the training data is the 10-year historical data from 1985 to 1994.

In each simulation process, there are two training methods for the simulated object. The first is to train a conditional probability table using the data in all the grids, and then use the conditional probability table to simulate the states of all grids. The conditional probability table obtained by this training method can reflect the numerical characteristic relationship between climate variables in the whole region. However, it can not show the distribution pattern of the characteristics of the simulated state in space. The second is to train the data in each grid separately. Because the state grading data in each grids is different, the conditional probability table of the simulated object trained in each grids is also different, and a total of $96 \times 48$ conditional probability tables are obtained. The conditional probability tables obtained by this training method can accurately reflect the different numerical characteristic relationship between the simulated object and the known object in different regions. However, due to the training of more conditional probability tables, the running time of this training method will be a little longer. Considering the great differences in the pattern of climate evolution in different regions, this paper uses the second data training

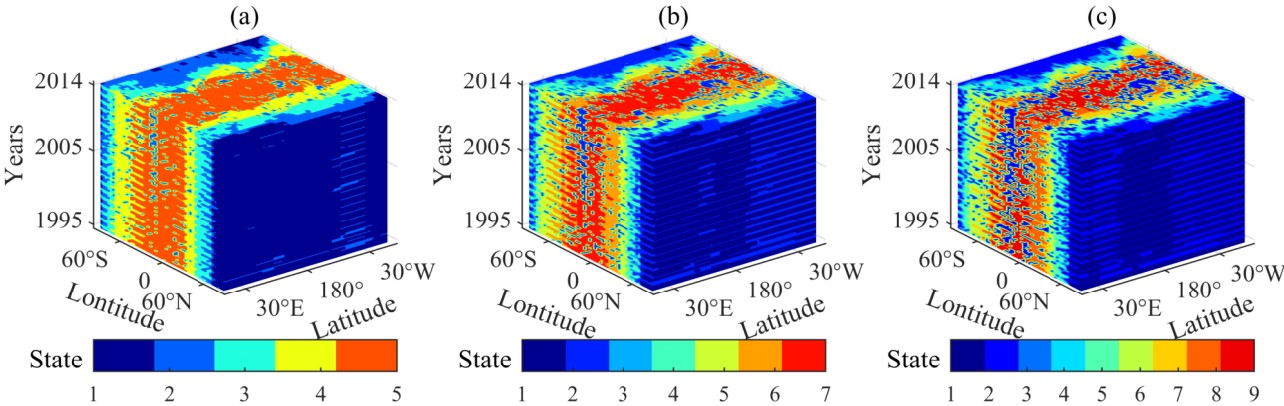

**Figure 3.** Climate state simulation results of the surface temperature by Bayesian Networks for 80 seasons (the period 1995-2014). (a) Categorized into 5 classifications; (b) Categorized into 7 classifications; (c) Categorized into 9 classifications.

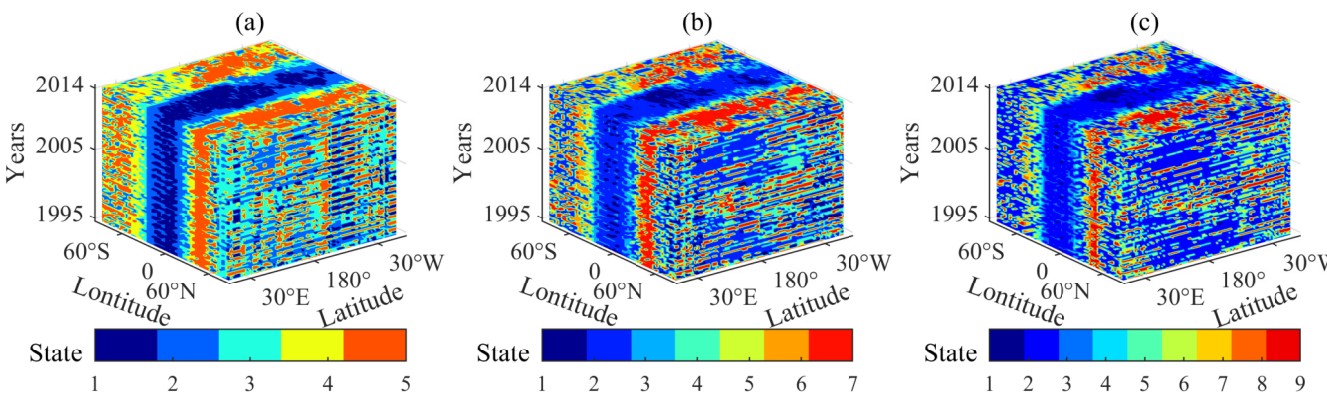

**Figure 4.** Climate state simulation results of the temperature of the atmosphere by Bayesian Networks for 80 seasons (the period 1995-2014). (a) Categorized into 5 classifications; (b) Categorized into 7 classifications; (c) Categorized into 9 classifications.

method in state simulation, which first divides the whole world into 96×48 grids, and then uses the data in each grids to train the conditional probability tables of the grid. After the training is completed, the data of the known climate variables will be used to simulate the unknown climate variables from 1995 to 2014. The simulation results are shown in Figure 3 and 4.

Figure 3 shows the climate state of the quarterly average of surface temperature from 1995-2014. The simulation results under different classifications all clearly show the global quarterly average surface temperature distribution with latitudinal variations. The surface temperature starts from the equator and decreases with the increase of latitude, so the temperature in the North and South Pole is the lowest. The climate state distribution of surface temperature is basically in line with the real world. Different from the simulation result of surface temperature, the quarterly average temperature of the atmosphere rises

from the equator and increases with the increase of latitude in Figure 4, which is also basically in line with the real world. The tropospheric height of the poles is lower and the tropospheric height of the equator is higher, and which phenomenon leads to the result that temperature of the troposphere at the same height is higher in the poles.

### 3.4 Local optimization

After the climate state simulation of Bayesian Networks, the numerical results simulated by the original GREB model are compared with the climate state results simulated by the Bayesian Networks, and the grids where the numerical result simulated by the original GREB model are not in the range of the climate state simulated by the Bayesian Networks are used as grids to be optimized. The GREB model uses the model code of the GREB in the Monash Simple Climate Model (MSCM) laboratory repository and runs the code in FORTRAN language.

Based on the optimized area of surface temperature simulations and temperature of the atmosphere obtained from the climate state accuracy comparison (see Appendix A for details), the original GREB model simulation results of the grid to be optimized in the optimized area are recalculated according to Eq. 5.

In terms of surface temperature simulation, the original GREB model at low latitudes shows high state accuracy, so a local optimization scheme is used, only for the middle and high latitudes. The empirical parameter for optimization range of surface temperature simulation has been determined to be 90°N to 30°N and 30°S to 90°S. The quarterly average surface average temperature simulated by the improved model for the period 1994-2015 are presented in Figure 5. In terms of temperature of the atmosphere simulation, a spatially global optimization approach has been chosen, owing to the higher global state accuracy of the Bayesian Networks. The quarterly average temperature of the atmosphere simulated by the improved model for the period 1994-2015 are presented in Figure 6. The details in Figure 5 and 6 show that the optimization data results are well characterized by localized abrupt changes, which means that the improved model are able to effectively address the inadequate response of the original GREB model to localized abrupt changes.

### 3.5 Evaluation of improved model

In order to evaluate the simulation accuracy of the improved model (the optimizated GREB model based on Bayesian Networks of climate state), the root mean square error (RMSE) between the simulated and actual values is defined to evaluate the model:

$$\text{RMSE} = \sqrt{\frac{1}{n} \sum \left(S_i - A_i\right)^2} \tag{12}$$

where $S_i$ represents the simulated value and $A_i$ represents the actual value, when analyzed spatially $n$ represents the length of time, whereas when analyzed temporally $n$ represents the number of grids in space. The accuracy of original GREB model simulation result was used as a comparison

The mean of RMSE between the simulated results of the original GREB model, as well as the improved models based on five, seven, and nine classifications, and the observed values for surface temperature were 13.26, 8.66, 8.85, and 9.81, respectively. For temperature of the atmosphere, the corresponding mean of RMSE were 72.19, 22.77, 20.12, and 17.76,

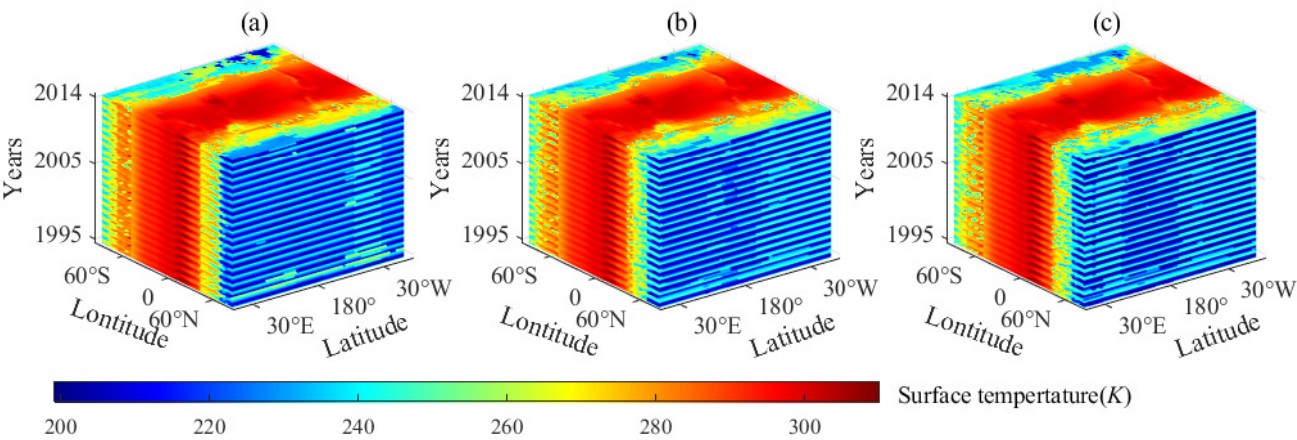

**Figure 5.** Quarterly average surface temperature for 80 seasons (the period 1995-2014). (a) Simulated by improved GREB model based on Bayesian Networks under 5 classifications; (b) Simulated by improved GREB model based on Bayesian Networks under 7 classifications; (c) Simulated by improved GREB model based on Bayesian Networks under 9 classifications.

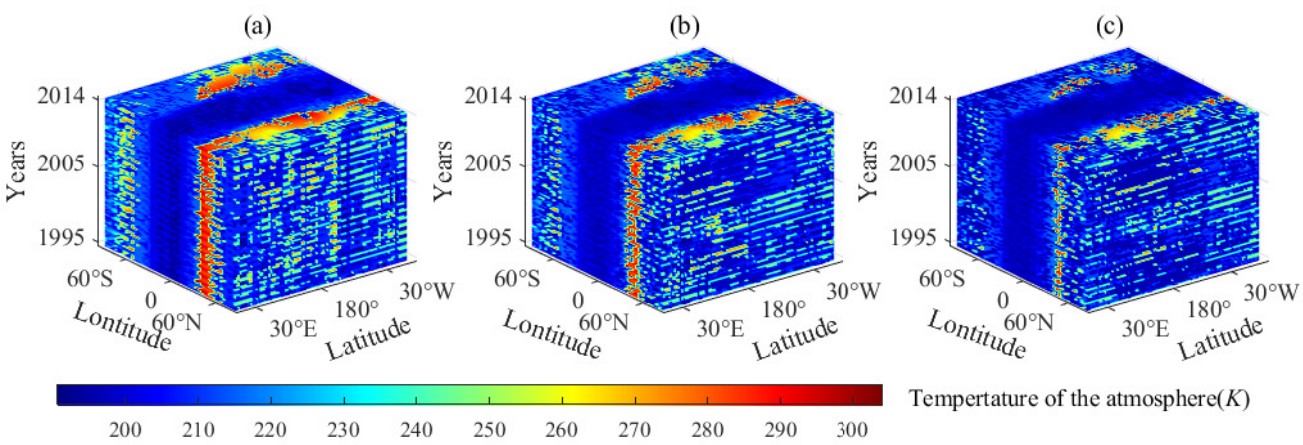

**Figure 6.** Quarterly average temperature of the atmosphere for 80 seasons (the period 1995-2014). (a) Simulated by improved GREB model based on Bayesian Networks under 5 classifications; (b) Simulated by improved GREB model based on Bayesian Networks under 7 classifications; (c) Simulated by improved GREB model based on Bayesian Networks under 9 classifications.

respectively. This result shows that the improved method significantly reduces the RMSE of the simulation, i.e., it improves the simulation accuracy. However, there are also significant differences between surface temperature and temperature of the atmosphere. There is no significant relationship between RMSE and classification in the simulation of surface temperature, while RMSE decreases with increasing classification in the simulation of temperature of the atmosphere.

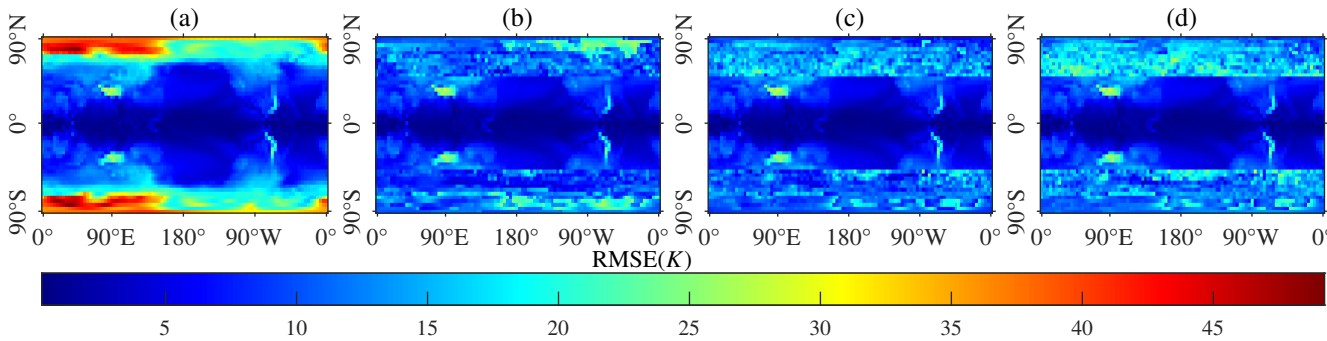

**Figure 7.** Spatial distribution of the RMSE for 80 seasons of the simulated surface temperature. (a) Simulated by original GREB model; (b) Simulated by optimizated GREB model based on Bayesian Networks under 7 classifications; (c) Simulated by optimized GREB model based on Bayesian Networks under 5 classifications; (d) Stimulated by optimized GREB model based on Bayesian Networks under 9 classifications.

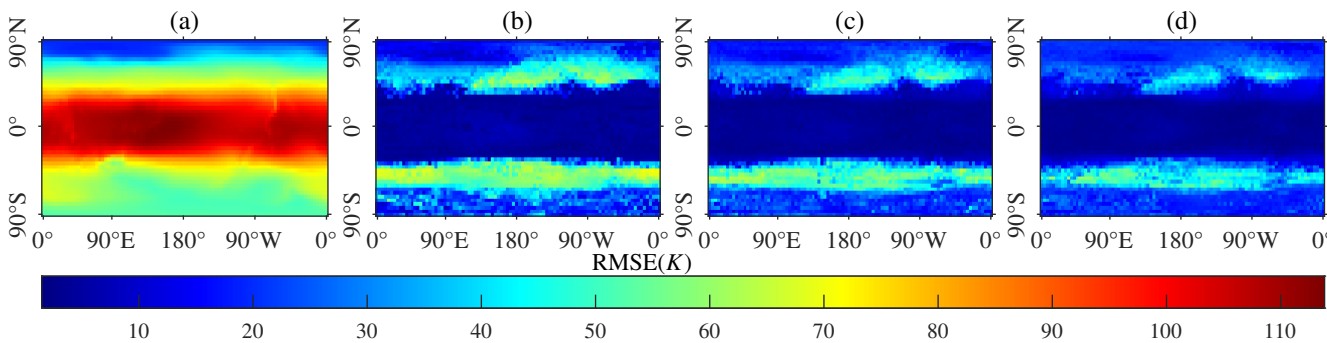

**Figure 8.** Spatial distribution of the RMSE for 80 seasons of the simulated temperature of the atmosphere. (a) Simulated by original GREB model; (b) Simulated by optimizated GREB model based on Bayesian Networks under 5 classifications; (c) Simulated by optimizated GREB model based on Bayesian Networks under 7 classifications; (d) Simulated by optimizated GREB model based on Bayesian Networks under 9 classifications.

Figure 7 depicts the spatial distribution of RMSE between the simulated surface temperature and the observed values for the original GREB model(Figure 7a) and the improved models based on five, seven, and nine classifications (Figure 7b-d). The comparison shows that the improved model significantly improves the simulation accuracy of the surface temperature in the polar regions. Figure 8 depicts the spatial distribution of RMSE between the simulated temperature of the atmosphere and the observed values for the original GREB model(Figure 8a) and the improved models based on five, seven, and nine classifications (Figure 8b-d). The comparison shows that the improved model significantly improves the simulation accuracy of temperature of the atmosphere at mid and low latitude regions. This is also well verified by the variation curve of RMSE along the latitude direction shown in Figure 9.

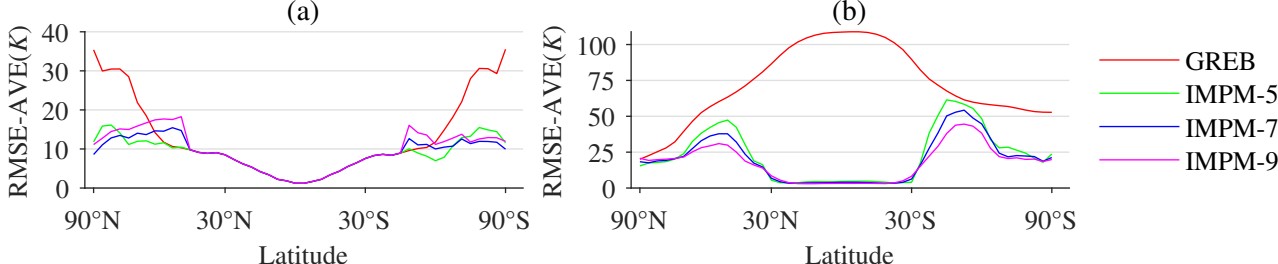

**Figure 9.** Mean of the spatial distribution of the RMSE in the latitudinal direction, GREB represents the results of the original GREB model simulation, IMPM − 5 represents the simulation results of the improved model under 5 classifications, IMPM − 7 represents the simulation results of the improved model under 7 classifications, IMPM − 9 represents the simulation results of the improved model under 9 classifications. (a) Comparison results of simulated temperature; (b) Comparison results of simulated temperature of the atmosphere.

Figure 10 shows the quarterly variability and trends of RMSE between 1995 and 2014 for both the surface temperature and temperature of the atmosphere. The comparison results demonstrate that the improved model significantly reduces the RMSE and exhibits temporal stability, indicating the robustness of the improved model. Moreover, the RMSE curves of the improved models exhibit the same seasonal cycle as the original GREB model, with the smallest RMSE occurring in the fourth quarter and the largest in the third quarter. This seasonal pattern can be attributed to the fact that the improved model is based on the modeling of the climate variable relationship within the GREB model, thus exhibiting similar temporal variation characteristics to those of the GREB model, which reflects the coarse-fine structure of improved model with the original GREB model as the global framework. The RMSE trends over time demonstrate that the improved model is temporally stable, and its accuracy does not deviate over time. This renders the improved model suitable for simulating surface temperature and temperature of the atmosphere over long time series. divergence

## 4 Conclusions and discussions

In this study, we introduced a coarse-fine structure to improve the GREB model based on Bayesian Networks. The improved model uses the GREB model as the basis of the global simulation framework and uses the Bayesian Networks to do local optimization. By introducing a Bayesian Networks, the results of the original GREB model are quickly evaluated with the climate state as the evaluation index, the local optimization region is confirmed, and the simulation results of the GREB model within the optimization region are recalculated, which makes the model accuracy significantly improved.

The improved model was evaluated by two cases: surface temperature and temperature of the atmosphere. The simulation results of the improved model show that the improved model has higher average accuracy and lower spatial variability compared to the original GREB model. This means that the improved model has better applicability and stability on a global scale. Meanwhile, on the time scale, the model maintains good robustness and does not suffer from the problem of accuracy diverges of traditional statistical models because the improved model uses the GREB model as the basic global framework. The results

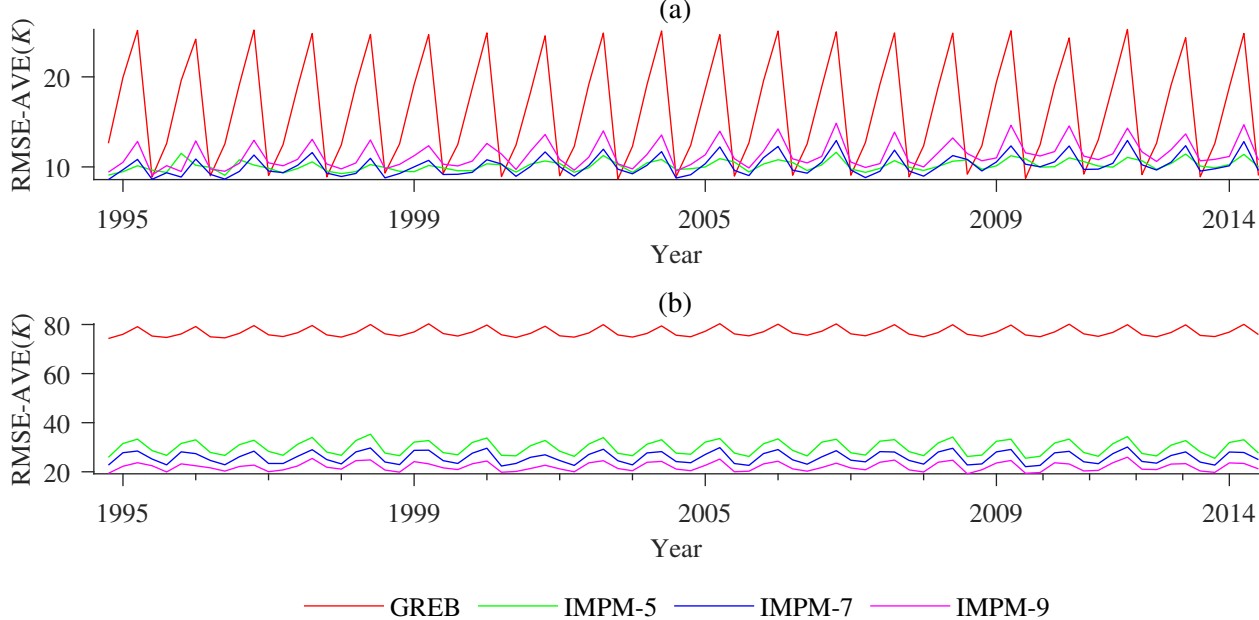

**Figure 10.** Quarterly variability and trends of RMSE between 1995 and 2014, GREB represents the results of the original GREB model simulation, IMPM − 5 represents the results of the improved model under 5 classifications, IMPM − 7 represents the results of the improved model under 7 classifications, IMPM − 9 represents the results of the improved model under 9 classifications. (a) Comparison results of simulated surface average temperature; (b) Comparison results of simulated atmospheric average temperature.

of the two study cases not only demonstrate that the improved method can also be used for the simulation of other climate variables within the GREB model. It also reveals the construction of coarse-fine models through a combination of dynamical and statistical methods as a potential means of improving climate simulation and prediction. This improved approach can overcome the shortcomings of a single dynamical model that cannot accurately describe many nonlinear processes in the climate system and can be applied to other dynamical models. In terms of development, the improved methods for improving
climate dynamical models by statistical methods show great possibilities for improving the accuracy of climate predictions.

In addition to the improved model with improved accuracy, the concept of evaluation through climate state introduced during the construction of the coarse-fine model can be well in studies on climate sensitivity(Dommenget, 2016; Kutzbach et al., 2013), extreme weather(Bellprat and Doblas-Reyes, 2016; Chen et al., 2018), and climate threshold(Mahlstein et al., 2015; Vogel et al., 2020), where the climate anomalies characterized by climate state can be a good indicator of climate change
trends.

However, this improved method still has some shortcomings.1) The scientific problem of categorization of climate variables attribute features. In this paper, the climate state of each variables is indicated by the classifications categorized by the Natural Breaks classification method according to the data characteristics and statistical regularities of the climate model, but this clas-

sification method changes with the data, and the data-based classification model may not be consistent with the actual climate evolution pattern. Therefore, the following studies can discuss related issues and choose the appropriate feature classification criteria to achieve a balance between different simulation. 2) Balance of accuracy and resolution. If the actual numeric rather than states are used as the calculation parameters, a higher resolution will be obtained, and of course the training data for each case will be reduced, which leads to the loss of accuracy. How to achieve the balance of accuracy and resolution will be an important issue. 3) Applicability of climate evolution models based on Bayesian Networks. Stable conditional probability tables can be trained with historical climate data to simulate climate state, but conditional probability tables cannot change over time and cannot be adapted to time-sensitive climate models. The following study can extend the applicability of the method by dynamically training Bayesian Networks on climate data.

*Code availability.*  The improved method in this paper was conducted in MATLAB R2021a. The code of the improved method used in this paper is archived on Zenodo(https://doi.org/10.5281/zenodo.7031997). The original GREB model uses the model code from the Monash Simple Climate Model (MSCM) laboratory repository for the GREB model and runs the code using the Fortran language. The model code is available from https://doi.org/10.5281/zenodo.2232282

*Data availability.*  The data used in this paper is is archived on Zenodo(https://doi.org/10.5281/zenodo.7031997). The data used for the analysis in this paper have been pre-processed and the original data can be gotten from Environmental Predic-tion(NCEP)/ National Center for Atmospheric Research(NCAR), download from https://psl.noaa.gov/data/gridded/data.ncep.reanalysis.pressure.html

**Appendix A:  State accuracy comparison**

In order to verify the reliability of the simulated climate state using the Bayesian Networks and to provide a basis for guiding the optimization of the GREB local simulation result, the state accuracy (dimensionless) was used to evaluate the reliability of the simulated climate state, which is expressed as:

$$\text{State accuracy} = \frac{n}{N} \tag{A1}$$

Where $n$ represents the number of time series in which the simulated state value of a grid is the same as the actual state value in the time series, in this case refers to the number of seasons; $N$ represents the total number of time series. State accuracy means the same proportion of simulated and actual states in the same grid.

The numerical results simulated by the original GREB model are also transformed into climate state by the natural breaks classification method for comparative evaluation. State accuracy averaged over space of different processes from 1985 to 1994 is shown in Figure A1, and the state accuracy of the surface temperature and the temperature of the atmosphere are shown in Figure A2 and A3. Overall, the comparison results (Figure A1) show that the Bayesian Networks has a higher simulation state accuracy in both surface temperature and the temperature of the atmosphere. This higher state accuracy indicates that

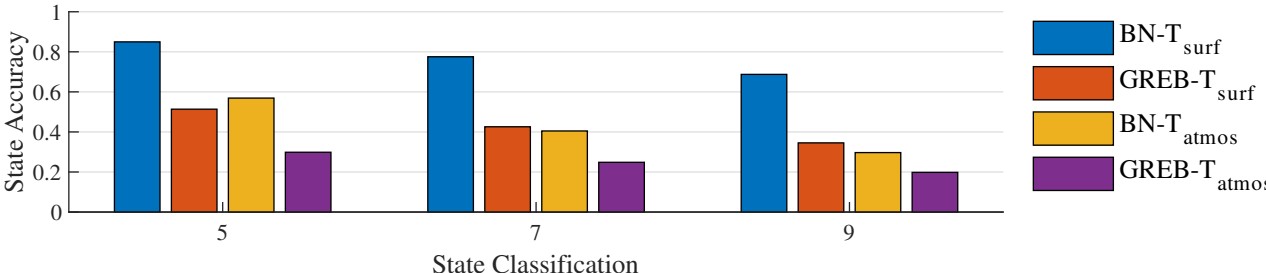

**Figure A1.** Comparison results of the mean of state accuracy of different methods. 5, 7, 9 are different classifications; $BN-T_{surf}$ represents the mean of state accuracy of surface temperature by Bayesian Networks simulation; $GREB-T_{surf}$ represents the mean of state accuracy of surface temperature by GREB model; $BN-T_{atmos}$ represents the mean of state accuracy of temperature of the atmosphere by Bayesian Networks simulation; $GREB-T_{atmos}$ represents the mean of state accuracy of temperature of the atmosphere by GREB model simulation.

the Bayesian Networks simulates climate state better than the GREB model, which provides a basis for evaluating the GREB model simulations with Bayesian Networks simulation results. Intrinsic to this result is the fact that since more observations
are involved in the simulation process (in the construction of conditional probability tables) in Bayesian Networks simulations, this allows the Bayesian Networks response to climate localized abrupt changes to be more pronounced.

When it comes to the number of classification, the total number of data remains unchanged, as the number of classification increases, the number of training data per classification decreases, it results in a decrease in the accuracy of the simulations of the two methods. This implies that the accuracy of the simulation can be stabilized at a high level when there is enough training
data in the long-period simulation.

Figures A2 and A3 elucidate the spatial distribution characteristics of the state accuracy of the Bayesian Networks simulation and that of the GREB model simulation, which provides a basis for the subsequent selection of regions for recalculating of GREB simulation data based on Bayesian Networks simulation result. The state accuracy of the Bayesian Networks simulation result is relatively uniform in spatial distribution and has no obvious spatial characteristics. However, the state accuracy of the
GREB model has obvious characteristics of latitude differentiation. Based on above, the state accuracy on the space is averaged along the latitudinal direction as shown in Figure A4. The variances of the state accuracy of Bayesian Networks simulation result in six cases are $0.016$ ($BN-5-T_{surf}$), $0.014$ ($BN-7-T_{surf}$), $0.014$ ($BN-9-T_{surf}$), $0.017$ ($BN-5-T_{atmos}$), $0.008$ ($BN-7-T_{atmos}$), $0.004$ ($BN-9-T_{atmos}$), and the variances of state accuracy of GREB model simulation result in six cases are $0.089$ ($GREB-5-T_{surf}$), $0.070$ ($GREB-7-T_{surf}$), $0.060$ ($GREB-7-T_{surf}$), $0.077$ ($GREB-5-T_{atmos}$), $0.054$
($GREB-7-T_{atmos}$), $0.036$ ($GREB-9-T_{atmos}$). The variance indicates that the fluctuation range of the state accuracy of the Bayesian Networks is much smaller than that of the GREB model along the latitude direction. This means that Bayesian Networks have a wide range of applications in global climate state simulation.

Although the Bayesian Networks has higher state accuracy in both simulations, we also found that the state simulation accuracy of the GREB model in the range of 30°S to 30°N tends to be higher than that of the Bayesian Networks when

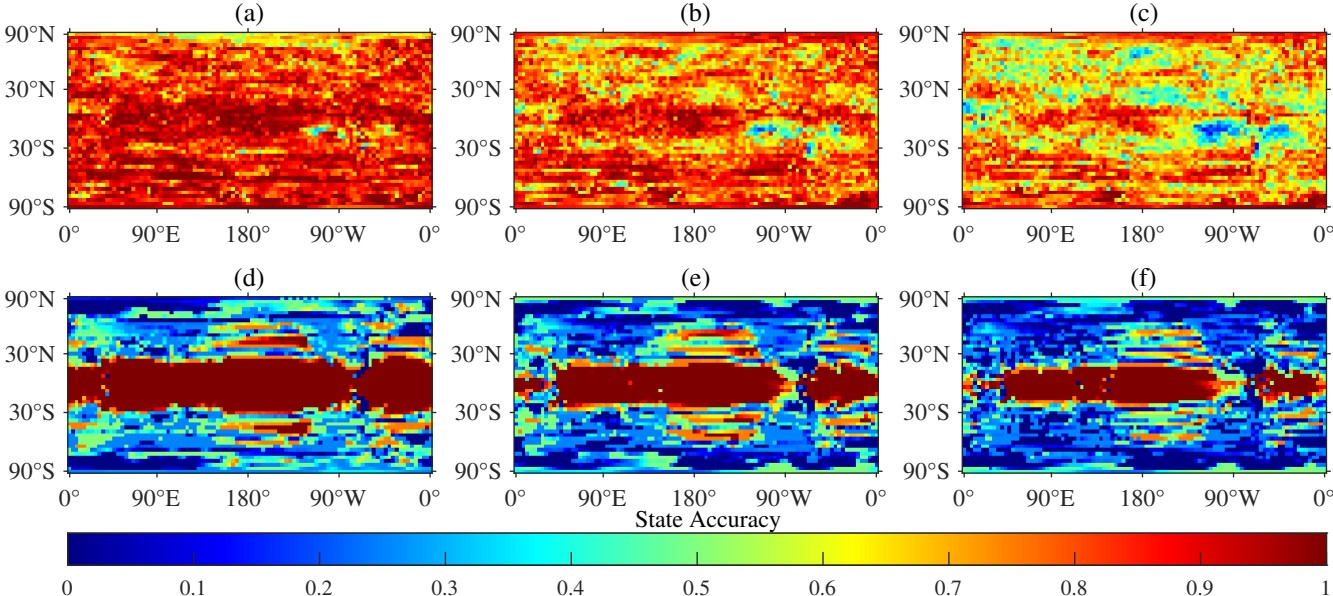

**Figure A2.** Comparison results of the state accuracy of simulation results of surface temperature. (a) State accuracy of Bayesian Networks under 5 classifications; (b) State accuracy of Bayesian Networks under 7 classifications; (c) State accuracy of Bayesian Networks under 9 classifications; (d) State accuracy of the GREB model under 5 classifications; (e) State accuracy of the GREB model under 7 classifications; (f) State accuracy of the GREB model under 9 classifications.

the classification numbers are 5, 7, and 9 in the surface temperature simulation. Therefore, we think that the GREB model can accurately represent the surface temperature simulation process in this range, and there is no abrupt change region that cannot be expressed, so in the subsequent optimization, only the range of 90°N to 30°N and 30°s to 90°N is selected as the optimization region for surface temperature simulation.

     Based on the above comparative analysis of the state accuracy of Bayesian Networks simulations and the state accuracy of
GREB model simulations, the range of 90°N to 30°N and 30°S to 90°S was selected as the empirical parameter for the range of subsequent data recalculating in surface temperature simulations, and the global range was selected as the range of data recalculating in temperature of the atmosphere simulations.

## Appendix B: Tables

*Author contributions.* ZL, ZZ, and WL designed the paper's ideas and methods. ZL, ZW, JW, implemented the method of the paper with
code. ZL, DL, and WL wrote the paper with considerable input from ZY and LY. ZW revised and checked the language of the paper.

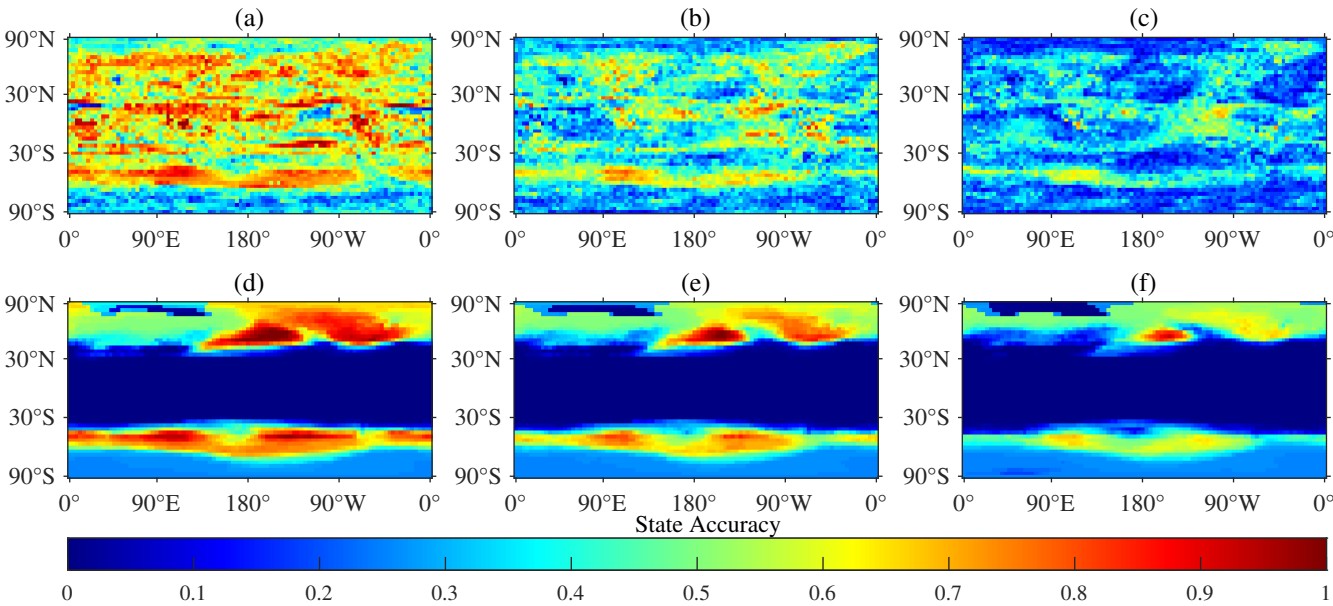

**Figure A3.** Comparison results of the state accuracy of simulation results of temperature of the atmosphere. (a) State accuracy of Bayesian Networks under 5 classifications; (b) State accuracy of Bayesian Networks under 7 classifications; (c) State accuracy of Bayesian Networks under 9 classifications; (d) State accuracy of the GREB model under 5 classifications; (e) State accuracy of the GREB model under 7 classifications; (f) State accuracy of the GREB model under 9 classifications.

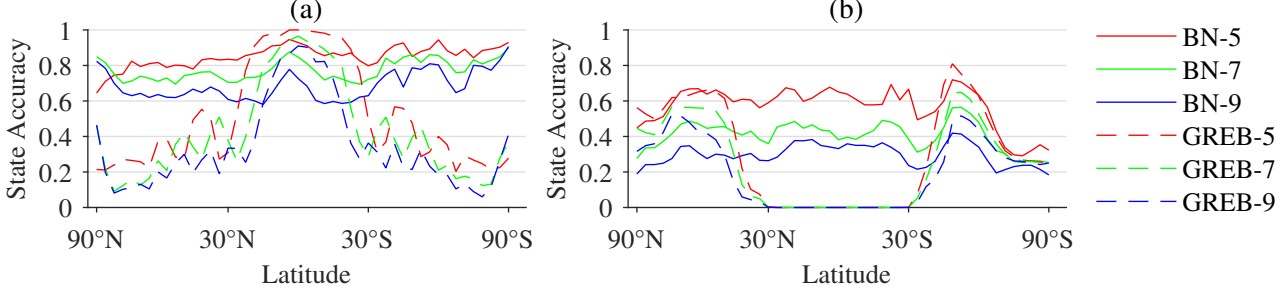

**Figure A4.** Mean of the spatial distribution of the state accuracy in the latitudinal direction, $BN-5$ represents the state accuracy of Bayesian Networks under state classification simulation, $BN-7$ represents the state accuracy of Bayesian Networks under 7 state classification simulation, $BN-9$ represents the state accuracy of Bayesian Networks under 9 state classification simulation, $GREB-5$ represents the state accuracy of the GREB model under 5 state classification simulation, $GREB-7$ represents the state accuracy of the GREB model under 7 state classification simulation, $GREB-9$ represents the state accuracy of the GREB model under 9 state classification simulation. (a) comparison results of surface temperature; (b) comparison results of temperature of the atmosphere.

**Table A1.** Five classification to indicate climate state

| State | $T_{surf}(K)$ | $T_{atmos}(K)$ | $F_{solar}$(W/m$^2$) | $T_{ocean}$(°C) | $q_air(kg/kg)$ | $CLD(kg/kg)$ | $\vec{u}(m/s)$ |
|---|---|---|---|---|---|---|---|
| 1 | <242.73 | <198.30 | <131.69 | <3.44 | <0.003 | <33.06 | <-3.44 |
| 2 | 242.73-264.81 | 198.29-205.11 | 131.69-160.62 | 3.44-10.82 | 0.003-0.007 | 33.06-45.51 | -3.44 - -0.9 |
| 3 | 264.81-279.03 | 205.11-212.03 | 160.62-193.67 | 10.82-17.86 | 0.007-0.011 | 45.51-56.24 | -0.90 - 1.51 |
| 4 | 279.03-291.24 | 212.03-217 | 193.67-224.48 | 17.86-24.17 | 0.011-0.016 | 56.24-66.1 | 1.51 - 4.55 |
| 5 | >291.24 | >217.00 | >224.48 | >24.17 | >0.016 | >66.10 | >4.55 |

**Table A2.** Seven classification to indicate climate state

| State | $T_{surf}(K)$ | $T_{atmos}(K)$ | $F_{solar}$(W/m$^2$) | $T_{ocean}$(°C) | $q_air(kg/kg)$ | $CLD(kg/kg)$ | $\vec{u}(m/s)$ |
|---|---|---|---|---|---|---|---|
| 1 | <236.92 | <196.02 | <122.28 | <1.97 | <0.003 | <28.24 | <-4.94 |
| 2 | 236.92-252.70 | 196.02-200.22 | 122.28-139.30 | 1.97-6.54 | 0.003-0.005 | 28.24-38.05 | -4.94 - -2.56 |
| 3 | 252.70-264.68 | 200.22-205.53 | 139.30-162.10 | 6.54-11.51 | 0.005-0.008 | 38.05-46.85 | -2.56 - -0.53 |
| 4 | 264.68-275.57 | 205.53-210.36 | 162.10-188.67 | 11.51-16.79 | 0.008-0.011 | 46.85-54.51 | -0.53-1.14 |
| 5 | 275.57-285.52 | >210.36-214.34 | 188.67-212.89 | >16.79-21.61 | 0.011-0.014 | >54.51-61.71 | 1.14-3.16 |
| 6 | 285.52-294.63 | >210.34-217.89 | 212.89-237.12 | >21.61-25.76 | 0.014-0.017 | >61.71-69.05 | 3.16-5.66 |
| 7 | >294.63 | >217.89 | >237.12 | >25.76 | >0.017 | >69.05 | >5.66 |

*Competing interests.* The authors declare that they have no conflict of interest.

*Acknowledgements.* This research has been supported by the National Natural Science Foundation of China (No.41976186, 42230406 and 42130103), the Postdoctoral Science Foundation of China (No. 2021M702757) and Postgraduate Research & Practice Innovation Program of Jiangsu Province(KYCX22-1578)

**Table A3.** Nine classification to indicate climate state

| State | $T_{surf}(K)$ | $T_{atmos}(K)$ | $F_{solar}$(W/m$^2$) | $T_{ocean}$(°C) | $q_a ir(kg/kg)$ | $CLD(kg/kg)$ | $\vec{u}(m/s)$ |
|---|---|---|---|---|---|---|---|
| 1 | <235.71 | <195.10 | <121.03 | <-0.67 | <0.001 | <25.31 | <-5.37 |
| 2 | 235.71-250.23 | 195.10-198.07 | 121.03-136.52 | -0.67-2.24 | 0.001-0.00 | 25.31-33.31 | -5.37- -3.41 |
| 3 | 250.23-259.75 | 198.07-201.69 | 136.52-154.71 | 2.24-6.58 | 0.003-0.005 | 33.31-40.68 | -3.41- -1.65 |
| 4 | 259.75-268.10 | 201.69-206.06 | 154.71-174.83 | 6.58-11.23 | 0.005-0.008 | 40.68-47.40 | -1.65 - -0.26 |
| 5 | 268.10-276.19 | >206.06-209.96 | 174.83-193.40 | 11.23-15.91 | 0.008-0.01 | 47.40-53.36 | -0.26-0.95 |
| 6 | 276.19-283.38 | >209.96-213.38 | 193.40-210.80 | 15.91-19.99 | 0.01-0.013 | 53.36-59.13 | 0.95-2.38 |
| 7 | 283.38-290.23 | >213.38-216.46 | 210.80-227.90 | 19.99-23.60 | 0.013-0.015 | 59.13-64.92 | 2.38-4.18 |
| 8 | 290.23-296.32 | >216.46-218.86 | 227.90-248.00 | 23.60-26.67 | 0.015-0.018 | 64.92-70.91 | 4.18-6.26 |
| 9 | >296.32 | >218.86 | >248.00 | >26.67 | >0.018 | >70.91 | >6.26 |

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
