# Peer review of "An improved method of the Globally Resolved Energy Balance Model by the Bayesian Networks"

_EGUsphere, 2022_

## Author Comment (AC4)

**Reviewer 1**

The manuscript "An improved method of the Globally Resolved Energy Balance Model by the Bayes network" by Liu et al. discusses a Bayes network approach to simulate global temperatures given some climate forcing boundary conditions such as solar radiation, cloud cover or water vapor. While the subject matter is interesting and should be considered for publication, the manuscript is largely unintelligible and should not get published. There are far too many aspects in this manuscript that would need substantial revisions. I recommend to reject this manuscript with an opportunity to resubmit a substantially revised analysis. Details below.
* * *
Major points:
(*) Clarity: The manuscript lags clarity on the methods, results and aims on many different levels. It is essentially unintelligible for the reader to understand what has been done, what the results are and how we did get there. Below are a few more points that may help the authors to improve on this.

(*) GREB model: The authors seem use the term GREB model as a general concept of how to simulate the global temperatues based on some boundary conditions (e.g. solar radiation). At the same time the term GREB model also refers to a model published by Dommenget et al.. This is confusing. It is unclear what the GREB model by Dommenget et al. has to do with the Bayes network approach the authors use. It seems they are essentially unrelated models.

**Response 1**
We utilize the trendcy equations of the different processes in the GREB model in the construction of the Bayes network, so it can be assumed that this Bayes network is abstracted through the GREB model. In our revised version, we further add the link between the improved model and the GREB model by using the GREB-based Bayes network as a local optimization of the GREB model, the main framework is shown below :

[Figure]

Fig. 1 Overall framework of the improved method

In the paper we have chosen the surface temperature as a case study to verify the reliability of this improved method, and therefore, the process of simulating the surface temperature in the GREB model is used as a proxy for the GREB model in the discussion in the paper.

(*) Language: The authors use terms that are not commonly used and are therefor hard to understand. Examples:

"the change law of climate system"

"the average accuracy variation trend chart"

**Response 2**

We have revised the language throughout the paper to bring it more in line with the expressions in the climate field.

(*) IMPM model: What is the IMPM model? It says it is "improved". Improved based on what? It seems to imply that it is based on the GREB model by Dommenget et al., but I do not see how. From the text it seems to have nothing to do with the GREB model by Dommenget et al..

**Response 3**

We further added the link between the improved model and the GREB model in revised version. we introduced a coarse-fine structure to improve the GREB model based on Bayes network. The improved model uses the GREB model as the basis of the global simulation framework and uses the Bayes network to do local optimization.

(*) "classification of climate elements is 5, 7, and 9": It is unclear what climate element classifications are. It is in particular unclear what a GREB model 5,7, and 9 would be in contrast to IMPM 5,7, and 9.

**Response 4**

The related concepts have been added in detail in the revised version. We divide the climate variable data (specific values) obtained from NCEP/NCAR into different classifications by classification, and these classifications is a indication of climate state.

5, 7, and 9 represent the three types of classification methods. GREB model 5,7, and 9 are contrast to IMPM 5,7, and 9. to discuss the reliability of the classification method.

(*) Accuracy: How is accuracy defined?

**Response 5**

We have added a specific definition of state accuracy in the paper, equation a1.

In order to verify the reliability of the simulated climate states using the Bayes network and to provide a basis for guiding the optimization of the GREB local simulation result, the state accuracy(dimensionless) was used to evaluate the reliability of the simulated climate states, which is expressed as:

$$\text{State accuracy} = \frac{n}{N}$$

Where $n$ represents the number of time series in which the simulated state value of a grid is the same as the actual state value in the time series, in this case refers to the number of seasons; $N$ represents the total number of time series. State accuracy means the same proportion of simulated and actual states in the same grid.

(*) "Nodes" vs. "classification of climate elements is 5, 7, and 9". The authors define several nodes in the GRBE model (.e.g. water vapor, winds, etc.) and they later discuss "classification of climate elements is 5, 7, and 9". Are these entirely unrelated concepts? It needs substantial revisions to explain this better.

**Response 6**

We have added further discussion in the paper.

Nodes is a graph or network concept, which in the analysis in the paper refers to a climate variable. 5, 7, and 9 refer to the simulated climate state, representing three experiments that are used to discuss the reliability of climate state classification.

In the original manuscript these concepts could be misunderstood due to excessive omissions, and in the revised version we have added the detail.

(*) Fig.2,3 : What is shown in this figure?  The figure is largely unclear. What are the colors? What are the units? What is shown on z-axis labeled "Seasons" 0 to 80? Does this figure has something to do with surface temperature [Celsius]?

 **Response 7**

We have added a more detailed description in the revised version.

Fig.2,3 shows the climate state of the quarterly average of surface temperature and temperature of the atmosphere from 1995-2014, which is simulated by bayes network. z-axis labeled has been revised to "years" 1995 to 2014. This figure represents the simulated climate state in which they are

located. In the case of surface temperature, for example, in the case of 5 state classification, the figure 1 represents a temperature less than 242.73 K, the figure 2 represents a temperature between 242.73 K and 264.91 K; and in the case of 7 state classification, the figure 1 represents a temperature less than 236.92 K in the case of category 7, the figure 2 represents a temperature between 236.92 K and 252.70 K. All classification intervals can be seen in Appendix B.

We chose group rather than specific values to represent these climate variables, the main starting point are to build coarse-fine structure which existence of different granularity and to improve the speed of the calculations so that they meet the original intention of the GREB model to be " a fast tool for the conceptual understanding and development of hypotheses for climate change studies"

(*) Fig.7: What is "accuracy variation trend"? It is largely unclear what "trend" could refer to.

**Response 8**

This was an inappropriate expression and we have changed it to " Mean of the spatial distribution of the state accuracy in the latitudinal direction" in the revised version.

In fact "trend" means spatial distribution of the state accuracy. Based on this comment, we have carefully revised the expression of the whole paper.

(*) GREB model surface temperature:   The GRBE model by Dommenget et al. is a flux corrected model that by construction has no biases (errors) in the simulation of the surface temperature. Then, how can the IMPM model be better at simulating surface temperature than the GRBE model, when the GREB model is by construction perfect?

**Response 9**

In a sense, the biases (errors) of the GREB model essentially comes from the construction perfect that applies to the overall global structure but makes it not respond well to local abrupt changes, while the IMPM (in the revised version) further improves the accuracy by using observational data and local optimization.
* * *
Other major points (as they appear in the text):
* * *
line 55 "... these two average state variables includes most of the climate processes of the GREB ..": Why? Why not water vapour? It is not obvious why water vapour would not be important.

**Response 10**

Of course, water vapor is an important variable in the GREB model (water vapor is one of the four prognostic variables in the GREB model). However, the purpose of choosing surface temperature and temperature of the atmosphere in the paper is to serve as a case study to verify the advantages and reliability of the improved method. We have revised expression in the paper.
* * *
line 189-191 "The tropospheric height of the poles is lower and the tropospheric height of the equator is higher, and which phenomenon leads to the result that temperature of the troposphere at the same height is higher in the poles.":
This statement appears to have nothing to do with the analysis presented. It certainly has nothing to do with the GREB model by Dommenget et al., as there is no such thing a   tropospheric height, as it is a one-layer model.
* * *
**Response 11**

This passage in the paper is some explanation of the characteristics of the simulation results, and it is mainly used to describe that the climate state simulated by Bayes networks is consistent with reality and is reliable. It is also to show that climate variables vary from region to region and therefore GREB's mean-state-based approach may introduce errors.

line 202 "The all average accuracy of different situations from 1985 to 1994 ...":   Does this imply the model is simulated for each year? Assuming they are different each year?
How would this work in the GREB model by Dommenget et al.? This model is not simulating internal variability, but only the response to changes in external boundary conditions.

**Response 12**

There is a misunderstanding that our model, like GREB, does not take into account the temporal variability, and if it did, it would have been trained in different periods.
The simulation is done for each season, not each year, which reflects seasonal variations. Here "average" means to average the state accuracy, which is used to prove that the Bayes network has better average accuracy. We analyze the average error on each year to show that the optimized model can effectively reflect the temporal variability of climate events due to the consideration of different climate modals.
* * *
Other minor points: not listed, as there are too many major points that need to be addressed first.

**Response 13**

In addition to the above targeted revisions, we added experiments to further reflect the improvement of the method on the GREB model (a coarse- fine structure improved model with the GREB model as the global framework and Bayes networks as the local optimization was constructed), and the language of the article was also embellished to make it more consistent with the expression of the climate field and to enrich the details in the paper.

---

## Author Comment (AC5)

**Reviewer 2**

In this study, the authors try to use the Bayes network method to construct a graph model, which is inspired from the structural relationship of a dynamic model named GREB. They find the graph model outperforms the GREB model. The methodology is sound and the results are interesting. However, this manuscript is not clear enough in terms of methods and results. Thus substantial revision is needed.

(1) The sensible heat expression (Qsense) in lines 65/72 and Equation (1 and 2) is inconsistent.

**Response 1**

We unified the representation of *Qsense* as $F_{sense}$ in the revised version of the article.

(2) Why the equations 3 and 4 can be derived from equations 1 and 2 need more explanations. What is the kinetic and thermodynamic basis for equations 1 and 2 that can be expressed using five variables (C, W, S, V, O)? For example, ocean temperature (O) is only controlled by radiation (S), and is independent of V. This may facilitate the construction of a graph network, but it is physically unreasonable. Please provide the physical basis for constructing the relationship (the directed edges).

**Response 2**

We have added further derivation details in the paper and modified the abbreviations of the variables from C, W, S, V, O to CLD, $q_{air}$, $F_{solar}$, $\vec{u}$, $T_{ocean}$ in the tendency equation, which makes the abstraction process more intuitive.

The nodes are selected considering not only the tendency equation but also the efficiency of the calculation, so the unselected nodes do not mean that they are unimportant in the physical, but the positive objects are discarded in the case of little impact on the accuracy of the results in order to improve the efficiency of the calculation and make it reach the starting point of GREB " a fast tool for the conceptual understanding and development of hypotheses for climate change studies"

(3) Line 151. What are the standards for data selection?

**Response 3**

We chose these data because they are needed in the simulations of the two cases chosen.

(4) The descriptions in Figures 2 and 3 are too sketchy. What are the units of the colors? What is the label of z-axis?

**Response 4**

Fig.2,3 shows the climate state of the quarterly average of surface temperature and temperature of

the atmosphere from 1995-2014, which is simulated by bayes network. z-axis labeled has been revised to "years" 1995 to 2014. This figure represent the climate state in which they are located. In the case of surface temperature, for example, in the case of 5 state classification, the figure 1 represents a temperature less than 242.73 K, the figure 2 represents a temperature between 242.73 K and 264.91 K; and in the case of 7 state classification, the figure 1 represents a temperature less than 236.92 K in the case of category 7, the figure 2 represents a temperature between 236.92 K and 252.70 K. All classification intervals can be seen in Appendix B.

We chose states rather than specific values to represent these climate variables, the main starting point being to improve the speed of the calculations so that they meet the original intention of the GREB model to be " a fast tool for the conceptual understanding and development of hypotheses for climate change studies"

(5) Please give more details about the natural break method.

**Response 5**

We have added a section in the methods section to give more details about the natural break method and why choose state as evaluation objects.

(6) Captions about the figures 5 and 6 maybe wrong. I guess the subplots (e) and (f) are related to the GREB results rather than IMPM.

**Response 6**

We have corrected the notes on Figures 5 and 6.

(7) The title is somewhat misleading. In fact, the authors refer to GREB equations to guide the construction of a graph model. After that, a completely new statistical model was evaluated using NCEP data. However, this statistical model is not used to optimize the GREB model in turn. In general, improving physical models using statistical models is achieved by optimizing uncertain empirical parameters. Therefore, it seems that the IMPM has nothing to do with the GREB improvement. The authors should reconsider whether the current title is accurate.

**Response7**

In response to this comment, we have added experiments to the paper. we introduced a coarse-fine structure to improve the GREB model based on Bayes network. The improved model uses the GREB model as the basis of the global simulation framework and uses the Bayes network (improved model in the original manuscript) to do local optimization. The concept of coarse-fine model provides a joint modeling approach of dynamical-statistical hybrid model that is different from the traditional use of statistical model to optimize the empirical parameters of the dynamical model.

**Response 8**

In addition to the above targeted revisions, we added experiments to further reflect the improvement of the method on the GREB model (a coarse- fine structure improved model with the GREB model as the global framework and Bayes networks as the local optimization was constructed, the main framework is shown below ), and the language of the article was also embellished to make it more consistent with the expression of the climate field and to enrich the details in the paper.

[Figure]

Fig. 1 Overall framework of the improved method

---

## Author Response (AR2)

Dear Editor and Reviewers:

This is a major reversion of our former manuscript "An improved method of the Globally Resolved Energy Balance Model by the Bayes network". Thank you for your interest and helpful comments on our paper. In the revised version, we reorganized our contents, added several important technological details, and extended the experiments and evaluations. The most significant differences of the reversion and original version are listed as following:

**Methods**:Based on the original method, we constructed an improved method based on coarse-fine structure with the GREB model as the global framework and Bayesian network as the local optimization through the coarse-fine framework, which further increases the connection between the improved model and the GREB model. To achieve this, we add the local optimization algorithm.

**Experiment**:In the case study section, we added experiments to further reflect the improvement of the method on the GREB model (a coarse-fine structure improved model with the GREB model as the global framework and Bayes networks as the local optimization was constructed). Comparative experiments on the numerical results of climate variables simulated by our new improved method validate its accuracy, applicability stability, and robustness. The original experiments on climate state simulation results are included as an appendix to demonstrate the advantages of the Bayesian network-based climate state simulation method for the selection of the optimization grid of new improved method.

For the color of the figures, we have check our figures using the Coblis to ensure that the colour schemes used in our figures allow readers with colour vision deficiencies to correctly interpret our findings.

To improve the language expressions, we have carefully checked and modified the manuscript accordingly, we also provide a detailed response as following. We hope this time our paper will meet the high standard criteria of the Geoscientific Model Development.

**Detailed response of reviewer's comments**

(Line and figure numbers refers to the line and figure numbers in the original manuscript)

**Reviewer 1**

**Comments 1:**

(*) GREB model: The authors seem use the term GREB model as a general concept of how to simulate the global temperatues based on some boundary conditions (e.g. solar radiation). At the same time the term GREB model also refers to a model published by Dommenget et al.. This is confusing. It is unclear what the GREB model by Dommenget et al. has to do with the Bayes network approach the authors use. It seems they are essentially unrelated models.

**Response 1:**

We utilize the trendcy equations of the different processes in the GREB model in the construction of the Bayes network, so it can be assumed that this Bayes network is abstracted through the GREB model. In our revised version, we further add the link between the improved model and the GREB model by using the GREB-based Bayes network as a local optimization of the GREB model, the main framework is shown below :

[Figure]

Fig. 1 Overall framework of the improved method

In the paper we have chosen the surface temperature as a case study to verify the reliability of this improved method, and therefore, the process of simulating the surface temperature in the GREB model is used as a proxy for the GREB model in the discussion in the paper.

**Comments 2:**

(*) Language: The authors use terms that are not commonly used and are therefor hard to understand.

Examples:

"the change law of climate system"

"the average accuracy variation trend chart"

**Response 2:**

We have revised the language throughout the paper to bring it more in line with the expressions in the climate field.

**Comments 3:**

(*) IMPM model: What is the IMPM model? It says it is "improved". Improved based on what? It seems to imply that it is based on the GREB model by Dommenget et al., but I do not see how. From the text it seems to have nothing to do with the GREB model by Dommenget et al..

**Response 3:**

We further added the link between the improved model and the GREB model in revised version. we introduced a coarse-fine structure to improve the GREB model based on Bayes network. The improved model uses the GREB model as the basis of the global simulation framework and uses the Bayes network to do local optimization.

**Comments 4:**

(*) "classification of climate elements is 5, 7, and 9": It is unclear what climate element classifications are. It is in particular unclear what a GREB model 5,7, and 9 would be in contrast to IMPM 5,7, and 9.

**Response 4:**

The related concepts have been added in detail in the revised version. We divide the climate variable data (specific values) obtained from NCEP/NCAR into different classifications by classification, and these classifications is a indication of climate state.

5, 7, and 9 represent the three types of classification methods. GREB model 5,7, and 9 are contrast to IMPM 5,7, and 9. to discuss the reliability of the classification method.

**Comments 5:**

(*) Accuracy: How is accuracy defined?

**Response 5:**

We have added a specific definition of state accuracy in the paper, equation a1.

In order to verify the reliability of the simulated climate states using the Bayes network and to provide a basis for guiding the optimization of the GREB local simulation result, the state accuracy(dimensionless) was used to evaluate the reliability of the simulated climate states, which is expressed as:

$$\text{State accuracy} = \frac{n}{N}$$

Where $n$ represents the number of time series in which the simulated state value of a grid is the same

as the actual state value in the time series, in this case refers to the number of seasons; $N$ represents the total number of time series. State accuracy means the same proportion of simulated and actual states in the same grid.

**Comments 6:**

(*) "Nodes" vs. "classification of climate elements is 5, 7, and 9". The authors define several nodes in the GRBE model (.e.g. water vapor, winds, etc.) and they later discuss "classification of climate elements is 5, 7, and 9". Are these entirely unrelated concepts? It needs substantial revisions to explain this better.

**Response 6:**

We have added further discussion in the paper.

Nodes is a graph or network concept, which in the analysis in the paper refers to a climate variable. 5, 7, and 9 refer to the simulated climate state, representing three experiments that are used to discuss the reliability of climate state classification.

In the original manuscript these concepts could be misunderstood due to excessive omissions, and in the revised version we have added the detail.

**Comments 7:**

(*) Fig.2,3 : What is shown in this figure?   The figure is largely unclear. What are the colors? What are the units? What is shown on z-axis labeled "Seasons" 0 to 80? Does this figure has something to do with surface temperature [Celsius]?

**Response 7:**

We have added a more detailed description in the revised version.

Fig.2,3 shows the climate state of the quarterly average of surface temperature and temperature of the atmosphere from 1995-2014, which is simulated by bayes network. z-axis labeled has been revised to "years" 1995 to 2014. This figure represents the simulated climate state in which they are located. In the case of surface temperature, for example, in the case of 5 state classification, the figure 1 represents a temperature less than 242.73 K, the figure 2 represents a temperature between 242.73 K and 264.91 K; and in the case of 7 state classification, the figure 1 represents a temperature less than 236.92 K in the case of category 7, the figure 2 represents a temperature between 236.92 K and 252.70 K. All classification intervals can be seen in Appendix B.

We chose group rather than specific values to represent these climate variables, the main starting point are to build coarse-fine structure which existence of different granularity and to improve the speed of the calculations so that they meet the original intention of the GREB model to be " a fast tool for the conceptual understanding and development of hypotheses for climate change studies"

**Comments 8:**

(*) Fig.7: What is "accuracy variation trend"? It is largely unclear what "trend" could refer to.

**Response 8:**

This was an inappropriate expression and we have changed it to " Mean of the spatial distribution

of the state accuracy in the latitudinal direction" in the revised version.

In fact "trend" means spatial distribution of the state accuracy. Based on this comment, we have carefully revised the expression of the whole paper.

**Comments 9:**

(*) GREB model surface temperature:   The GRBE model by Dommenget et al. is a flux corrected model that by construction has no biases (errors) in the simulation of the surface temperature. Then, how can the IMPM model be better at simulating surface temperature than the GRBE model, when the GREB model is by construction perfect?

**Response 9:**

In a sense, the biases (errors) of the GREB model essentially comes from the construction perfect that applies to the overall global structure but makes it not respond well to local abrupt changes, while the IMPM (in the revised version) further improves the accuracy by using observational data and local optimization.

**Comments 10:**

line 55 "... these two average state variables includes most of the climate processes of the GREB ..": Why? Why not water vapour? It is not obvious why water vapour would not be important.

**Response 10:**

Of course, water vapor is an important variable in the GREB model (water vapor is one of the four prognostic variables in the GREB model). However, the purpose of choosing surface temperature and temperature of the atmosphere in the paper is to serve as a case study to verify the advantages and reliability of the improved method. We have revised expression in the paper.

**Comments 11:**

line 189-191 "The tropospheric height of the poles is lower and the tropospheric height of the equator is higher, and which phenomenon leads to the result that temperature of the troposphere at the same height is higher in the poles.":

This statement appears to have nothing to do with the analysis presented. It certainly has nothing to do with the GREB model by Dommenget et al., as there is no such thing a  tropospheric height, as it is a one-layer model.

**Response 11:**

This passage in the paper is some explanation of the characteristics of the simulation results, and it is mainly used to describe that the climate state simulated by Bayes networks is consistent with reality and is reliable. It is also to show that climate variables vary from region to region and therefore GREB's mean-state-based approach may introduce errors.

**Comments 12:**

line 202 "The all average accuracy of different situations from 1985 to 1994 ...":   Does this imply

the model is simulated for each year? Assuming they are different each year?

How would this work in the GREB model by Dommenget et al.? This model is not simulating internal variability, but only the response to changes in external boundary conditions.

**Response 12:**

There is a misunderstanding that our model, like GREB, does not take into account the temporal variability, and if it did, it would have been trained in different periods.

The simulation is done for each season, not each year, which reflects seasonal variations. Here "average" means to average the state accuracy, which is used to prove that the Bayes network has better average accuracy. We analyze the average error on each year to show that the optimized model can effectively reflect the temporal variability of climate events due to the consideration of different climate modals.

**Reviewer 2**

**Comments 1:**

(1) The sensible heat expression (Qsense) in lines 65/72 and Equation (1 and 2) is inconsistent.

**Response 1:**
We unified the representation of *Qsense* as $F_{sense}$ in the revised version of the article.

**Comments 2:**

(2) Why the equations 3 and 4 can be derived from equations 1 and 2 need more explanations. What is the kinetic and thermodynamic basis for equations 1 and 2 that can be expressed using five variables (C, W, S, V, O)? For example, ocean temperature (O) is only controlled by radiation (S), and is independent of V. This may facilitate the construction of a graph network, but it is physically unreasonable. Please provide the physical basis for constructing the relationship (the directed edges).

**Response 2:**
We have added further derivation details in the paper and modified the abbreviations of the variables from C, W, S, V, O to CLD, $q_{air}$, $F_{solar}$, $\vec{u}$, $T_{ocean}$ in the tendency equation, which makes the abstraction process more intuitive.

The nodes are selected considering not only the tendency equation but also the efficiency of the calculation, so the unselected nodes do not mean that they are unimportant in the physical, but the positive objects are discarded in the case of little impact on the accuracy of the results in order to improve the efficiency of the calculation and make it reach the starting point of GREB " a fast tool for the conceptual understanding and development of hypotheses for climate change studies"

**Comments 3:**

(3) Line 151. What are the standards for data selection?

**Response 3:**
We chose these data because they are needed in the simulations of the two cases chosen.

**Comments 4:**

(4) The descriptions in Figures 2 and 3 are too sketchy. What are the units of the colors? What is the label of z-axis?

**Response 4:**
Fig.2,3 shows the climate state of the quarterly average of surface temperature and temperature of the atmosphere from 1995-2014, which is simulated by bayes network. z-axis labeled has been revised to "years" 1995 to 2014. This figure represent the climate state in which they are located. In the case of surface temperature, for example, in the case of 5 state classification, the figure 1 represents a temperature less than 242.73 K, the figure 2 represents a temperature between 242.73 K and 264.91 K; and in the case of 7 state classification, the figure 1 represents a temperature less than 236.92 K in the case of category 7, the figure 2 represents a temperature between 236.92 K and 252.70 K. All classification intervals can be seen in Appendix B.

We chose states rather than specific values to represent these climate variables, the main starting point being to improve the speed of the calculations so that they meet the original intention of the GREB model to be " a fast tool for the conceptual understanding and development of hypotheses for climate change studies"

**Comments 5:**

(5) Please give more details about the natural break method.

**Response 5:**
We have added a section in the methods section to give more details about the natural break method and why choose state as evaluation objects.

**Comments 6:**

(6) Captions about the figures 5 and 6 maybe wrong. I guess the subplots (e) and (f) are related to the GREB results rather than IMPM.

**Response 6:**
We have corrected the notes on Figures 5 and 6.

**Comments 7:**

(7) The title is somewhat misleading. In fact, the authors refer to GREB equations to guide the construction of a graph model. After that, a completely new statistical model was evaluated using NCEP data. However, this statistical model is not used to optimize the GREB model in turn. In general, improving physical models using statistical models is achieved by optimizing uncertain empirical parameters. Therefore, it seems that the IMPM has nothing to do with the GREB

improvement. The authors should reconsider whether the current title is accurate.

**Response 7:**

In response to this comment, we have added experiments to the paper. we introduced a coarse-fine structure to improve the GREB model based on Bayes network. The improved model uses the GREB model as the basis of the global simulation framework and uses the Bayes network (improved model in the original manuscript) to do local optimization. The concept of coarse-fine model provides a joint modeling approach of dynamical-statistical hybrid model that is different from the traditional use of statistical model to optimize the empirical parameters of the dynamical model.